# What Does Thompson Sampling Optimize?

Yanlin Qu [1]   Hongseok Namkoong [1]   Assaf Zeevi [1]

## Abstract

Thompson Sampling is one of the most widely used and studied bandit algorithms, known for its simple structure, low regret performance, and solid theoretical guarantees. Yet, in stark contrast to most other families of bandit algorithms, the exact mechanism through which posterior sampling (as introduced by Thompson) is able to "properly" balance exploration and exploitation, remains a mystery. In this paper, we show that the core insight to address this question stems from recasting Thompson Sampling as an online optimization algorithm. To distill this, we introduce a time invariant notion of regret that summarizes cumulative regret across horizons (through a regret bound), leading to a time invariant Bellman-optimal policy. It turns out that Thompson Sampling admits an online optimization form that mimics the structure of the Bellman-optimal policy, where greediness is regularized by a measure of residual uncertainty. When viewed through this new lens of online optimization, Thompson Sampling can be understood and improved in a principled manner, by comparing it against the Bellman-optimal benchmark.

## 1. Introduction

**Background and motivation.** Thompson Sampling is a heuristic algorithm, introduced by (Thompson, 1933) in the context of solving treatment allocation in medical trials; the objective is to maximize patient outcomes while simultaneously learning the best treatment. This motivating application has since been abstracted to what we recognize today as the multi-armed bandit (MAB) problem (Robbins, 1952). The algorithm proceeds in each round to sample from the posterior distribution, the updated belief over problem parameters, and then select the treatment (arm) that is perceived to be optimal in the sampled environment.

[1]Decision, Risk, and Operations Division, Columbia Business School, New York, United States. Correspondence to: Yanlin Qu <qu.yanlin@columbia.edu>.

*Proceedings of the 43$^{rd}$ International Conference on Machine Learning*, Seoul, South Korea. PMLR 306, 2026. Copyright 2026 by the author(s).

Despite its conceptual elegance, Thompson Sampling remained relatively obscure for decades. Early research in bandit problems focused more on asymptotic regret analysis, such as the fundamental performance bounds in (Lai & Robbins, 1985), as well as on the development of non-randomized strategies, including the Upper Confidence Bound (UCB) algorithm proposed by (Auer et al., 2002a). Interest in Thompson Sampling surged after its remarkably strong empirical performance was demonstrated in several studies, including (Scott, 2010; Chapelle & Li, 2011), highlighting the potential of this randomized approach. Since then, practitioners have applied Thompson Sampling across a wide range of domains, including online advertising (e.g., (Agarwal, 2013)), recommendation systems (e.g., (Kawale et al., 2015)), and website optimization (e.g., (Hill et al., 2017)). Meanwhile, a substantial body of theoretical work has been developed to bound the regret of Thompson Sampling, including frequentist regret bounds in (Agrawal & Goyal, 2012; 2013) and Bayesian regret bounds in (Russo & Van Roy, 2014b; 2016).

Although the above mentioned bounds place the heuristic algorithm on more rigorous footing, their proofs do not fully elucidate why and in what exact manner does the simple structure of Thompson Sampling lead to the regret performance investigated in the aforementioned studies. The challenge, at least in part, lies in the fact that the structure of Thompson Sampling has little, if anything, to do with explicit regret optimization, or principled explore-exploit considerations. In this paper we aim to shed some light on this elusive topic.

**Design principles for MAB algorithms.** In contrast to Thompson Sampling, most other MAB-type algorithms are designed with a clear focus on regret minimization. In the frequentist setting, the storied UCB algorithm creates a simple index rule based on the principle of optimism in the face of uncertainty. Specifically, a scaled confidence interval (CI), anchored at the empirical mean reward, is computed for each arm, and the highest value of the upper confidence bound serves as the index for arm selection in each round. The intuition here is fairly clear: the upper confidence bounds serves to encode the uncertainty associated with the empirical mean estimator and hence *regularize* the "greediness" of the former. The regret analysis reveals that the scaling factor in the CI must grow logarithmically in the

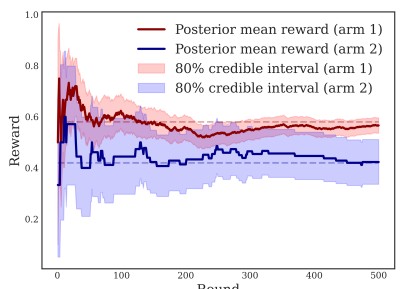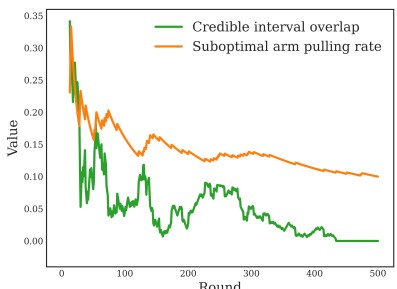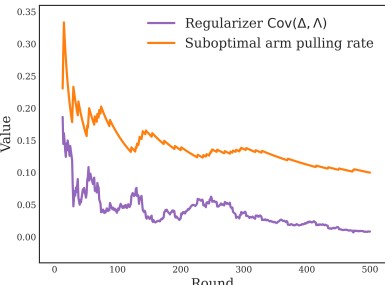

*Figure 1.* Thompson Sampling plays a Bernoulli bandit. Left: Dynamics of credible intervals. Middle: Credible interval overlap as an intuitive uncertainty measure. Right: The uncertainty measure that guides the exploration of Thompson Sampling.

number of rounds to balance the two competing terms in the regret bound that can be attributed to *exploration* and *exploitation*. In this manner, we gain a clear understanding of why UCB works.

In the adversarial setting where the reward sequence is controlled by an adversary, most widely studied algorithms are predicated on rather explicit optimization principles. For example, the celebrated EXP3 algorithm (Auer et al., 2002b), and more broadly the family of HEDGE algorithms, can be viewed as a form of *online mirror descent* that, at each round, maximizes an estimated reward regularized by an entropy penalty term; see Chapters 5.4 and 6.2 of (Hazan et al., 2016) for details. In short, both UCB as well as the EXP3 family of algorithms exhibit transparent design principles, as their key components (the logarithmic multiplier in UCB and the multiplicative update rule in EXP3) arise from explicit optimization formulations.

In the Bayesian setting, the pioneering work of (Gittins, 1979) approaches the algorithm design question using classical dynamic programming principles (Bellman, 1957) that focus on the expected cumulative discounted reward. The UCB principle, with its clear focus on regret minimization, has also been "exported" to the Bayesian setting; see, for example, BayesUCB (Kaufmann et al., 2012). The essential idea is to substitute the notion of *credible intervals* for that of CIs. The advantage of this modification is that a 90% credible interval contains the true mean reward with *exactly* 90% probability, whereas the coverage of the analogous 90% confidence interval depends on the quality of the normal approximation. There are various other instances of Bayesian bandit algorithms that perform explicit optimization at each round, a notable example is Information-Directed Sampling (IDS) (Russo & Van Roy, 2014a).

**The key research question and contribution of this paper.** In stark contrast to the algorithms surveyed above, Thompson Sampling makes decisions by simply drawing a *single* sample from each posterior distribution. Unlike EXP3 and IDS, no optimization is performed during its execution, and in contrast to UCB and BayesUCB, no approximate regret

bound is optimized in its design. Despite the absence of an explicit optimization formulation, Thompson Sampling has consistently demonstrated strong empirical performance and enjoys solid theoretical guarantees. This naturally raises a fundamental question: *What objective, exactly, is Thompson Sampling optimizing?* More specifically, does Thompson Sampling have some explicit regret minimization mechanism buried within its deceptively simple appearance?

In this paper, we aim to harness dynamic programming principles to shed light on the optimization considerations underlying Thompson Sampling. Different from Gittins' discounted reward formulation, we introduce a *squared regret* formulation, which is more aligned with the goal of long-term regret minimization. In this new formulation, Thompson Sampling has an online optimization form, outlined by the new Bellman equation. At each round, this simple randomized rule turns out to minimize a suitable notion of instantaneous regret, regularized adaptively by an uncertainty measure based on point-biserial correlation. We illustrate the role of this regularizer in Figure 1. Recall that Thompson Sampling leverages uncertainty to regularize greediness in a probabilistic manner: the higher the probability that the current leader is not truly optimal, the more frequently the other arm is sampled. This probability is intuitively reflected by the overlap of credible intervals (left plot), but not entirely, since the credible intervals eventually detach but the exploration continues (middle plot). This issue is resolved by the regularizer in our online optimization formulation, which quantifies the exploratory effect indefinitely (right plot).

Through the online optimization lens, we understand Thompson Sampling in a more principled and less heuristic manner: its exploration is no longer just an ad-hoc byproduct of posterior sampling, but rather can be understood as the result of an explicit trade-off between instantaneous regret and remaining uncertainty that regularizes the former, akin to what we are familiar with in other MAB-type algorithms. Moreover, there is a Bellman-optimal benchmark in the new formulation that allows us to improve Thompson Sampling with a compass in hand.

## 2. Preliminaries

### 2.1. Bayesian stochastic bandits

For simplicity of exposition, and to stay true to Thompson's original 1933 setup, we consider a two-armed Bayesian stochastic bandit. (The principles and key ideas carry over to the $K$-armed case, as discussed before Theorem 4.2.) The two arms are labeled with 1 and 2. Their joint reward distribution $P_\theta$ depends on an (unknown) environment parameter $\theta \in \Theta$. Before the game begins, $\theta$ is drawn from a prior distribution $\pi_0$ and remains fixed throughout the game. At each round, a potential reward vector is drawn independently from $P_\theta$, but only the entry corresponding to the pulled arm is observed. Conditional on $\theta$, these potential reward vectors form an independent and identically distributed (iid) sequence

$$R_0|\theta, R_1|\theta, \ldots \overset{\text{iid}}{\sim} P_\theta, \quad \theta \sim \pi_0.$$

After $t$ rounds, each involving a partial observation of a potential reward vector, the posterior distribution $\pi_t$ of $\theta$ is obtained by updating $\pi_0$ according to Bayes' rule. For simplicity, we take the environment parameter to be the mean reward vector

$$\mathbb{E}[R_0|\theta] = \mathbb{E}[(R_{1,0}, R_{2,0})|\theta] = (\theta_1, \theta_2) = \theta.$$

To make the next decision, Thompson Sampling draws $\theta'$ from $\pi_t$ and selects arm $A_t = \operatorname{argmax}(\theta'_1, \theta'_2)$ as if $\theta'$ were the true mean reward vector. After pulling the selected arm, based on the reward observed $R_{A_t,t}$, the belief is updated from $\pi_t$ to $\pi_{t+1}$, and the process repeats.

### 2.2. An MDP view

As noted by (Gittins, 1979), the Bayesian stochastic bandit can be viewed as a Markov decision process (MDP); see page 31 of (Ghavamzadeh et al., 2015) for an illustrative example. The MDP formulation is as follows:

- State: current belief $\pi_t$.

- Action: arm pulled $A_t$.

- Transition: updating $\pi_t$ to $\pi_{t+1}$ after observing the $A_t$-th entry of $R_t \sim P_{\theta''}$ where $\theta'' \sim \pi_t$.

- MDP reward: expected next-round reward $\mathbb{E}_{\pi_t} R_{A_t,t}$.

Note that the next potential reward vector is drawn from the posterior predictive distribution, so the system can evolve forward without knowing which $\theta$ was drawn and fixed at the beginning. This MDP view provides a natural framework for analyzing Bayesian bandit algorithms directly, without resorting to frequentist analysis followed by integration over the prior. We adopt this view in our analysis.

Viewing the Bayesian stochastic bandit as an MDP, Thompson Sampling induces a Markov chain on the space of beliefs, as its transition from $\pi_t$ to $\pi_{t+1}$ only depends on a sample from the posterior ($\theta' \sim \pi_t$) and a sample from the posterior predictive ($R_t \sim P_{\theta''}$, $\theta'' \sim \pi_t$). Note that, unlike algorithms such as the Gittins index (Gittins, 1979), Thompson Sampling does not attempt to solve the MDP in any dynamic programming sense. Its actions are guided entirely by posterior sampling, without computing value functions or optimizing over future trajectories.

## 3. Squared Regret Formulation

Given the MDP, to formulate an optimization perspective, we must first identify a well-defined objective. In the bandit literature, algorithmic performance is typically evaluated in terms of regret, which is the difference between the reward actually received and the best possible reward that could have been obtained in hindsight. Consider a generic stationary Markov policy $Q$, which maps the current belief $\pi_t$ to a pulling probability vector $q_t$. After $t$ rounds, $Q$ will pull arm $k$ in the next round with probability $q_{k,t}$, i.e., $A_t|\pi_t \sim q_t$. The expected next-round regret is

$$r(q_t; \pi_t) = \mathbb{E}_{\pi_t} \left[\max(\theta_1, \theta_2) - \theta_{A_t}\right].$$

The cumulative (Bayesian) regret up to time $T$ is

$$
\begin{aligned}
\mathcal{R}_T(Q; \pi_0) =& \mathbb{E}_{\pi_0} \left[\sum_{t=0}^{T-1} \left[\max(\theta_1, \theta_2) - \theta_{A_t}\right]\right] \\
=& \mathbb{E}_{\pi_0} \left[\sum_{t=0}^{T-1} \mathbb{E}_{\pi_0} \left[\max(\theta_1, \theta_2) - \theta_{A_t}\Big|\pi_t\right]\right] \\
=& \mathbb{E}_{\pi_0} \left[\sum_{t=0}^{T-1} r(q_t; \pi_t)\right].
\end{aligned}
$$

The ultimate goal of bandit algorithm design is to minimize cumulative regret $\mathcal{R}_T(Q; \pi_0)$ across horizons $T$. To summarize this diverging sequence as a single finite quantity, we define (cumulative) squared regret

$$\mathcal{R}^2(Q; \pi_0) = \mathbb{E}_{\pi_0} \left[\sum_{t=0}^{\infty} r^2(q_t; \pi_t)\right].$$

Note that $\mathcal{R}^2(Q; \pi_0)$ controls $\{\mathcal{R}_T(Q; \pi_0) : T \geq 1\}$ via

$$
\begin{aligned}
\mathcal{R}_T(Q; \pi_0) \leq& \mathbb{E}_{\pi_0} \left[\left(\sum_{t=0}^{T-1} 1\right)^{1/2} \left(\sum_{t=0}^{T-1} r^2(q_t; \pi_t)\right)^{1/2}\right] \\
\leq& \sqrt{T} \cdot \left(\mathbb{E}_{\pi_0} \left[\sum_{t=0}^{T-1} r^2(q_t; \pi_t)\right]\right)^{1/2} \\
\leq& \sqrt{\mathcal{R}^2(Q; \pi_0) \cdot T},
\end{aligned}
$$

$$(1)$$

where Cauchy's inequality and Jensen's inequality are used. As the pre-multiplier in the regret bound (1), the squared regret is a natural and meaningful objective to minimize. In particular, Thompson Sampling $Q^{\text{TS}}$ achieves finite squared regret, which is a direct corollary of the information-theoretic analysis in (Russo & Van Roy, 2016).

**Proposition 3.1** ($\mathcal{R}^2$-finiteness of $Q^{\text{TS}}$). *If there exists a finite constant $\sigma > 0$ such that the posterior predictive distribution of the reward ($R_t \sim P_{\theta''}, \theta'' \sim \pi_t$) is always $\sigma$-sub-Gaussian, then Thompson Sampling satisfies $\mathcal{R}^2(Q^{\text{TS}}; \pi_0) < \infty$, and hence $\mathcal{R}_T(Q^{\text{TS}}; \pi_0) = O(\sqrt{T})$.*

This shows that the set of $\mathcal{R}^2$-finite policies is non-empty. Thanks to the regret bound (1), each policy in this set achieves $O(\sqrt{T})$ cumulative regret, which is minimax optimal (Auer et al., 2002b). These observations make the $\mathcal{R}^2$-optimal policy $Q^{\text{R2}} = \operatorname{argmin}_Q \mathcal{R}^2(Q; \cdot)$ worth studying, as it not only achieves $O(\sqrt{T})$ cumulative regret but also attains the best possible constant in the regret bound (1). More importantly, studying the $\mathcal{R}^2$-optimal policy may shed light on the optimization considerations underlying Thompson Sampling, as $Q^{\text{R2}}$ achieves what $Q^{\text{TS}}$ achieves ($\mathcal{R}^2$-finiteness) but entirely through principled optimization.

## 4. Online Optimization Form

### 4.1. Bellman equation

To minimize the squared regret, we derive the corresponding Bellman equation. Let $V(\pi_t) = \mathcal{R}^2(Q^{\text{R2}}; \pi_t)$ be the minimal squared regret incurred from $\pi_t$ onward, achieved by the $\mathcal{R}^2$-optimal policy. This value function satisfies

$$V(\pi_t) = \min_{q_t} \left[ r^2(q_t; \pi_t) + q_t \cdot \mathbb{E}_{\pi_t}\left[V(\pi_{t+1})|A_t = \cdot\right] \right]. \tag{2}$$

Basically, the current $V$-value equals the instantaneous squared regret plus the expected future $V$-value, minimized over all possible values of the pulling probability vector $q_t$. According to the Bellman equation (2), $Q^{\text{R2}}$ is given by

$$q_t^{\text{R2}} = \operatorname*{argmin}_{q_t} \left[ r^2(q_t; \pi_t) + q_t \cdot \mathbb{E}_{\pi_t}\left[V(\pi_{t+1})|A_t = \cdot\right] \right].$$

If we choose the expected next-round reward

$$x_t = q_t \cdot \mathbb{E}_{\pi_t}\theta = q_{1,t}\mathbb{E}_{\pi_t}\theta_1 + q_{2,t}\mathbb{E}_{\pi_t}\theta_2$$

as the decision variable, then $Q^{\text{R2}}$ reveals an intuitive online optimization form

$$x_t^{\text{R2}} = \operatorname*{argmin}_{x_t} \left[ \left(\mathbb{E}_{\pi_t}\max(\theta_1, \theta_2) - x_t\right)^2 + \nu^{\text{R2}}(\pi_t)x_t \right], \tag{3}$$

where the *regularizer* $\nu^{\text{R2}}(\pi_t)$ is given by

$$\left[ \frac{\mathbb{E}_{\pi_t}\left[V(\pi_{t+1})|A_t = 1\right] - \mathbb{E}_{\pi_t}\left[V(\pi_{t+1})|A_t = 2\right]}{\mathbb{E}_{\pi_t}\theta_1 - \mathbb{E}_{\pi_t}\theta_2} \right]_+. \tag{4}$$

The online objective in (3) consists of two terms: an instantaneous loss term for exploitation (decreasing in $x_t$) and a linear regularization term for exploration (increasing in $x_t$). The greediness that would result from minimizing the first term alone is regularized by the second term.

*Remark* 4.1. We take the positive part $(\cdot)_+ = \max(\cdot, 0)$ when defining $\nu^{\text{R2}}(\pi_t)$ in (4), because it is clear that $x_t^{\text{R2}} = \max(\mathbb{E}_{\pi_t}\theta_1, \mathbb{E}_{\pi_t}\theta_2)$ whenever $\nu^{\text{R2}}(\pi_t) \leq 0$.

### 4.2. Tension measure

In the online optimization form (3), the regularizer $\nu^{\text{R2}}(\pi_t)$ determines how much exploitation should be traded off against exploration, quantifying the current tension between exploration and exploitation. To be specific, in (4), the first (respectively, second) term in the numerator represents the future squared regret after pulling arm 1 (respectively, arm 2). If the numerator is positive (respectively, negative), then arm 2 (respectively, arm 1) favors exploration, as pulling it leads to lower future squared regret. If the denominator is positive (respectively, negative), then arm 1 (respectively, arm 2) favors exploitation, as pulling it leads to higher immediate mean reward. As a result, when the ratio is positive, there is a clear tension between exploration and exploitation. Quantifying this tension requires looking into the far future, as computing $\nu^{\text{R2}}(\pi_t)$ (to implement $Q^{\text{R2}}$) requires solving the Bellman equation (2).

### 4.3. Thompson Sampling

Instead of the $\mathcal{R}^2$-optimal policy itself, perhaps its online optimization form (3) is more valuable. This form reveals what a reasonable bandit algorithm (that achieves $O(\sqrt{T})$ cumulative regret) should look like when the MAB problem is viewed through the lens of online optimization: minimizing the instantaneous squared regret with some linear regularization. We now show that Thompson Sampling also takes this form, focusing on the two-armed case

$$q_t^{\text{TS}} = (P_{\pi_t}(\theta_1 > \theta_2), P_{\pi_t}(\theta_1 \leq \theta_2)).$$

The corresponding expected next-round reward is

$$x_t^{\text{TS}} = P_{\pi_t}(\theta_1 > \theta_2)\mathbb{E}_{\pi_t}\theta_1 + P_{\pi_t}(\theta_1 \leq \theta_2)\mathbb{E}_{\pi_t}\theta_2. \tag{5}$$

In fact, for Thompson Sampling, the $K$-armed case can be viewed as repeating the two-armed case $K$ times to determine the $K$ pulling probabilities. For example, the probability of pulling arm 1

$$q_{1,t}^{\text{TS}} = P_{\pi_t}(\theta_1 > \theta_2, \ldots, \theta_K) = P_{\pi_t}(\theta_1 > \theta_{-1})$$

is determined by hedging between two random variables: $\theta_1$ and $\theta_{-1} = \max\{\theta_2, \ldots, \theta_K\}$. In this sense, (Thompson, 1933) focus on *two samples* already contains the core idea of Thompson Sampling as a multi-armed bandit algorithm. This idea now admits an online optimization form.

**Theorem 4.2** (Online optimization). *The online optimization form of Thompson Sampling is*

$$x_t^{\mathrm{TS}} = \underset{x_t}{\operatorname{argmin}} \left[ (\mathbb{E}_{\pi_t} \max(\theta_1, \theta_2) - x_t)^2 + \nu^{\mathrm{TS}}(\pi_t) x_t \right],$$

*where* $\nu^{\mathrm{TS}}(\pi_t) = \operatorname{Cov}_{\pi_t}(\theta_1 - \theta_2, \operatorname{sign}(\theta_1 - \theta_2))$.

All proofs are in Appendix A. According to Thompson's original idea, posterior sampling corresponds to the expected next-round reward $x_t^{\mathrm{TS}}$ given by (5). This $x_t^{\mathrm{TS}}$ now emerges from the above online optimization form, which is identical to (3) except for a different regularizer. The Bellman-optimal regularizer $\nu^{\mathrm{R2}}(\pi_t)$ is replaced by a simpler one $\nu^{\mathrm{TS}}(\pi_t)$, which turns out to be a covariance.

In this online optimization framework, different policies are fully characterized by their regularizers, so policy design becomes regularizer engineering. In particular, the randomized behavior of Thompson Sampling now admits a clean variational description, which allows us to improve it by modifying its regularizer. A natural first attempt is to multiply the regularizer by a constant, but this modification results in a failure mode known as *incomplete learning*.

**Proposition 4.3** (Incomplete learning). *For each* $\lambda \neq 1$, *there exists a prior under which the policy*

$$x_t^\lambda = \underset{x_t}{\operatorname{argmin}} \left[ (\mathbb{E}_{\pi_t} \max(\theta_1, \theta_2) - x_t)^2 + \lambda \nu^{\mathrm{TS}}(\pi_t) x_t \right]$$

*suffers from incomplete learning, i.e., it fully commits to one arm while the alternative may be the best arm.*

This result highlights that regularizer engineering is a delicate task. To make meaningful modifications that improve Thompson Sampling, we need principled guidance on what constitutes a better regularizer. In this regard, the Bellman-optimal regularizer $\nu^{\mathrm{R2}}(\pi_t)$ provides exactly such a compass, as $Q^{\mathrm{R2}}$ attains the best possible constant in the regret bound (1). By comparing the two regularizers $\nu^{\mathrm{TS}}(\pi_t)$ and $\nu^{\mathrm{R2}}(\pi_t)$ both analytically and numerically, we devote the rest of this paper to identifying and addressing the issues of Thompson Sampling in a principled manner.

### 4.4. Uncertainty measure

In Theorem 4.2, the regularizer of Thompson Sampling $\nu^{\mathrm{TS}}(\pi_t)$ is the covariance between the following two fundamental quantities:

$$\Delta = \theta_1 - \theta_2 \quad \text{the reward gap between the two arms,}$$
$$\Lambda = \operatorname{sign}(\theta_1 - \theta_2) \quad \text{the identity of the optimal arm.}$$

The study of the relationship between a metric (continuous) variable and a dichotomous (binary) variable dates back to (Pearson, 1909), and their "biserial" covariance admits a well-known expression; see, e.g., (Lev, 1949).

**Proposition 4.4** (Covariance factorization). *If* $\operatorname{Var}_{\pi_t} \Lambda = 0$, *then* $\operatorname{Cov}_{\pi_t}(\Delta, \Lambda) = 0$. *Otherwise,*

$$\frac{\operatorname{Cov}_{\pi_t}(\Delta, \Lambda)}{\operatorname{Var}_{\pi_t} \Lambda} = \frac{\mathbb{E}_{\pi_t}[\Delta | \Delta > 0] + \mathbb{E}_{\pi_t}[-\Delta | \Delta \leq 0]}{2}.$$

The covariance $\operatorname{Cov}_{\pi_t}(\Delta, \Lambda)$ is the product of two terms: a variance and the average of two expectations. The variance of the identity of the optimal arm $\operatorname{Var}_{\pi_t} \Lambda$ measures the remaining uncertainty about which arm is better. The two expectations $\mathbb{E}_{\pi_t}[\Delta | \Delta > 0]$ and $\mathbb{E}_{\pi_t}[-\Delta | \Delta \leq 0]$ correspond to the expected regret incurred by pulling arm 2 when arm 1 is optimal, and by pulling arm 1 when arm 2 is optimal. Their average can be viewed as a notion of instantaneous regret. Taken together, the regularizer of Thompson Sampling $\nu^{\mathrm{TS}}(\pi_t)$ is a regret-scaled uncertainty measure.

Figure 1 in the introduction demonstrates the ability of $\operatorname{Cov}_{\pi_t}(\Delta, \Lambda)$ to effectively capture remaining uncertainty in the bandit setting. As Thompson Sampling plays a two-armed Bernoulli bandit, we contrast the rate at which the suboptimal arm is pulled, with the extent of credible interval overlap (middle plot); and then with the biserial covariance (right plot). The overlap of credible intervals is an intuitive way to encode uncertainty (left plot), in similar fashion to the frequentist overlap of confidence intervals: the higher the overlap the more Thompson Sampling should be allocating arm pulls towards exploration to resolve said uncertainty. This gives rise to an increased frequency of sampling from the suboptimal arm, as evident in the middle plot. The right plot depicts that the qualitative behavior of this "exploration rate" is well captured by the biserial covariance. This connection is not a coincidence: the biserial covariance, the quantitative notion of uncertainty, temporally correlates strongly (a Pearson correlation of 0.995) with the more qualitative notion of overlap of credible intervals.

### 4.5. The two regularizers

Recall that $\nu^{\mathrm{R2}}(\pi_t)$ measures the current tension between exploration and exploitation, so the $\mathcal{R}^2$-optimal policy pulls the runner-up arm more often when doing so yields greater future benefit. In contrast, $\nu^{\mathrm{TS}}(\pi_t)$ measures the remaining uncertainty about which arm is better, so Thompson Sampling pulls the runner-up arm more often when the identity of the optimal arm is less clear. This comparison reveals that the motivation behind Thompson Sampling to pull the runner-up arm is not entirely compelling, as doing so does not necessarily make the situation clearer, especially when the leading arm is more uncertain than the runner-up arm. For example, when $\pi_t = N(1, 100) \times N(-1, 0.01)$, arm 1 favors both exploration (learning more) and exploitation (earning more). In this case, there is uncertainty but little tension, i.e., $\nu^{\mathrm{TS}}(\pi_t) \gg \nu^{\mathrm{R2}}(\pi_t)$, so Thompson Sampling pulls arm 2 more frequently than necessary.

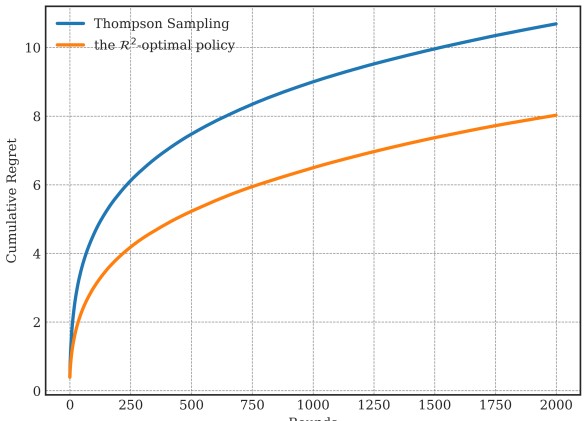 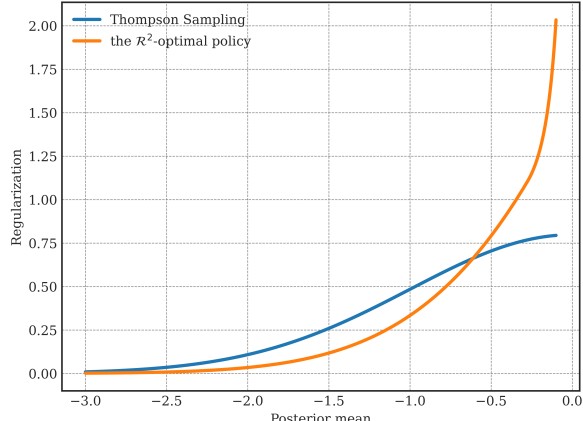

**Figure 2.** Thompson Sampling and the $\mathcal{R}^2$-optimal policy play a Gaussian bandit with reward variance 1. Left: Comparing their cumulative regret $\mathcal{R}_T(Q^{\text{TS}}; \pi_0)$ vs. $\mathcal{R}_T(Q^{\text{R2}}; \pi_0)$ where $\pi_0 = N(0,1) \times \delta_0$ (20K trials). Right: Comparing their regularizers $\nu^{\text{TS}}(N(\mu, 1) \times \delta_0)$ vs. $\nu^{\text{R2}}(N(\mu, 1) \times \delta_0)$ where $\mu \in [-3, 0]$.

### 4.6. Information-Directed Sampling

Before making further comparisons in the next section, we briefly discuss Information-Directed Sampling (IDS) (Russo & Van Roy, 2014a), another notable member of the family of $\mathcal{R}^2$-finite policies. After the change of variables from $q_t$ to $x_t$, IDS is given by

$$x_t^{\text{IDS}} = \underset{x_t}{\operatorname{argmin}} \frac{\bar{r}^2(x_t; \pi_t)}{\mathcal{I}(x_t; \pi_t)},$$

where $\bar{r}(x_t; \pi_t) = r(q_t; \pi_t)$ and $\mathcal{I}(x_t; \pi_t)$ is a notion of information gain. It is straightforward to rewrite IDS in the online optimization form (3), and the corresponding regularizer is

$$\nu^{\text{IDS}}(\pi_t) = \frac{\mathcal{I}(\mathbb{E}_{\pi_t}\theta_2; \pi_t) - \mathcal{I}(\mathbb{E}_{\pi_t}\theta_1; \pi_t)}{\mathbb{E}_{\pi_t}\theta_1 - \mathbb{E}_{\pi_t}\theta_2} \cdot \min_{x_t} \frac{\bar{r}^2(x_t; \pi_t)}{\mathcal{I}(x_t; \pi_t)}.$$

The second term is the minimal "information ratio", while the first term, similar to $\nu^{\text{R2}}(\pi_t)$ given by (4), is a tension measure. This term is positive when one arm gives more reward while the other arm gives more information (i.e., when there is a clear tension between exploration and exploitation). The structural similarity between $\nu^{\text{IDS}}(\pi_t)$ and $\nu^{\text{R2}}(\pi_t)$ may help explain why IDS outperforms Thompson Sampling, as the former is "closer" to the Bellman-optimal benchmark than the latter.

## 5. Thompson vs. Bellman

To benchmark Thompson Sampling, we take a closer look at the $\mathcal{R}^2$-optimal policy, presenting a closed-form solution (to the Bellman equation (2)) in the one-armed case and an approximate implementation in the two-armed case. Comparing the two regularizers allows us to identify and address the issues of Thompson Sampling, underscoring the appeal of a principled framework with a well-defined benchmark.

### 5.1. One arm

In the one-armed case, where one of the two arms is fully known, the $\mathcal{R}^2$-optimal policy turns out to be fully tractable, i.e., the Bellman equation (2) can be solved in closed form. Without loss of generality, we take arm 2 to be the known arm, with $\theta_2 \equiv 0$ (i.e., $\theta_2 \sim \delta_0$).

**Proposition 5.1** (Closed-form solution). *When $\theta_2 \equiv 0$ and $\mathbb{E}_{\pi_t}\theta_1 \neq 0$, the $\mathcal{R}^2$-optimal policy pulls the unknown arm 1 with probability*

$$q_{1,t}^{\text{R2}} = \min\left(\frac{\mathbb{E}_{\pi_t}[(\theta_1)_+]}{|\mathbb{E}_{\pi_t}\theta_1|}, 1\right).$$

In Figure 2, we compare Thompson Sampling with the $\mathcal{R}^2$-optimal policy in the one-armed case. The left panel of Figure 2 shows that the $\mathcal{R}^2$-optimal policy achieves substantially lower cumulative regret than Thompson Sampling, confirming the benefit of minimizing the regret bound (1). The reason behind the gap can be understood via regularizer comparison, thanks to the shared online optimization form. In the right panel of Figure 2, we compare the two regularizers along the prior sequence $\pi_0(\mu) = N(\mu, 1) \times \delta_0$, with $\mu \in [-3, 0]$, to illustrate how the two policies respond as the cost of exploration decreases. As the posterior mean reward of the unknown arm 1 increases from $-3$ to $0$, the cost of exploration decreases from 3 to 0. In particular, when $\mu = 0$ (so the two arms have the same posterior mean reward), the exploration of arm 1 becomes cost-free and should therefore be carried out with probability one. The right panel of Figure 2 shows that the regularizer of the $\mathcal{R}^2$-optimal policy does explode, thereby encouraging pure exploration (i.e., pulling arm 1 with probability one). In contrast, the regularizer of Thompson Sampling grows gradually, indicating that it does not fully account for the vanishing cost of exploration, as it is fundamentally an uncertainty measure.

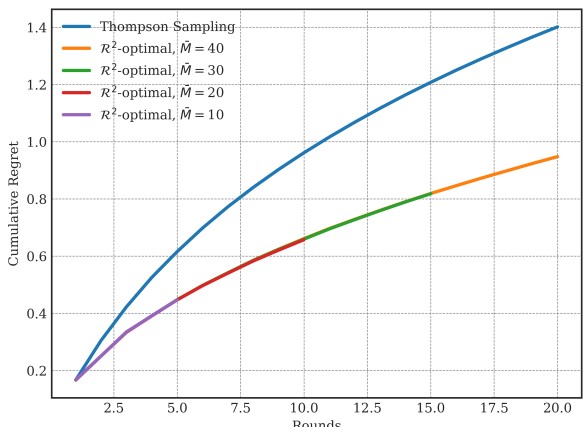 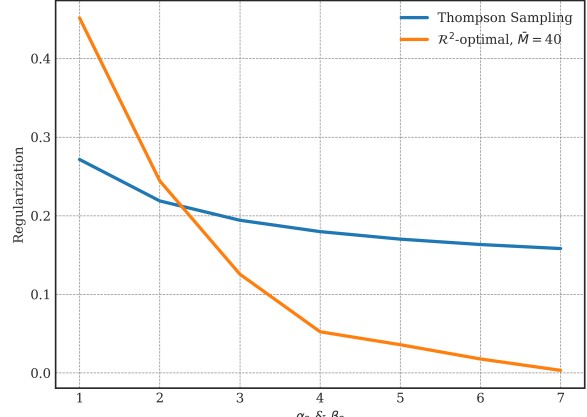

*Figure 3.* Thompson Sampling and the $\mathcal{R}^2$-optimal policy (with different values of the approximation parameter $\bar{M}$) play a Bernoulli bandit. Left: Comparing their cumulative regret $\mathcal{R}_T(Q^{\text{TS}}; \pi_0)$ vs. $\mathcal{R}_T(Q^{\text{R2}}; \pi_0)$ where $\pi_0 = \text{Beta}(1,1) \times \text{Beta}(1,1)$ (200K trials). Right: Comparing their regularizers $\nu^{\text{TS}}(\text{Beta}(5,4) \times \text{Beta}(k,k))$ vs. $\nu^{\text{R2}}(\text{Beta}(5,4) \times \text{Beta}(k,k))$ where $k = 1, \ldots, 7$.

Before turning to the two-armed case, we make an interesting observation about pure exploration. In Proposition 5.1, note that the condition $\mathbb{E}_{\pi_t}[(\theta_1)_+] + \mathbb{E}_{\pi_t}\theta_1 = 0$ marks a phase transition between randomized ($q_{1,t}^{\text{R2}} < 1$) and deterministic ($q_{1,t}^{\text{R2}} = 1$) behavior. In the Gaussian case, this phase transition can be characterized explicitly.

**Proposition 5.2** (Phase transition). *When $\theta_2 \equiv 0$ and $\theta_1 \sim N(\mu_t, \sigma_t^2)$ under $\pi_t$,*

$$q_{1,t}^{\text{R2}} = 1 \quad \Leftrightarrow \quad \mu_t/\sigma_t \geq \bar{x},$$

*where $\bar{x} \approx -0.276$ is the unique root of the increasing function $x\Phi(x) + \phi(x) + x$.*

That is, the unknown arm 1 is pulled with probability one if and only if its signal-to-noise ratio exceeds $-0.276$. The value $0.276$ can be interpreted as the relative price Bellman is willing to pay for the exploratory benefit of pulling the unknown arm 1. When the cost of exploration falls below this threshold (e.g., $-\mu_t/\sigma_t = 0.2 < 0.276$), Bellman pulls the unknown arm 1 with probability one to fully capitalize on the available arbitrage. This quantitative result further illustrates the value of introducing Bellman's principle into the MAB problem through our squared regret formulation.

*Remark* 5.3. If we look closely at the right panel of Figure 2, we can spot the phase transition of the $\mathcal{R}^2$-optimal policy at $-0.276$, where the regularizer curve becomes slightly less smooth than else where.

### 5.2. Two arms

In the two-armed case, the Bellman equation (2) can no longer be solved in closed form, but we manage to approximately implement the $\mathcal{R}^2$-optimal policy for Bernoulli bandits. Before using it to benchmark Thompson Sampling, we briefly discuss the approximation method. The implementation details are in Appendix B.

In the Beta–Bernoulli setting, the reward distribution is Bernoulli, while both the prior and posterior are Beta, so the belief space is parametrized by two pairs of positive integers

$$\{\text{Beta}(\alpha_1, \beta_1) \times \text{Beta}(\alpha_2, \beta_2) : \alpha_1, \beta_1, \alpha_2, \beta_2 \geq 1\}.$$

Let $V_{\alpha_1, \beta_1, \alpha_2, \beta_2} = V(\text{Beta}(\alpha_1, \beta_1) \times \text{Beta}(\alpha_2, \beta_2))$ be the solution to the Bellman equation (2). The expected $V$-value reduction after pulling arm 1, denoted by $V'_{\alpha_1, \beta_1, \alpha_2, \beta_2}$, is given by

$$V_{\alpha_1, \beta_1, \alpha_2, \beta_2} - \frac{\alpha_1 V_{\alpha_1+1, \beta_1, \alpha_2, \beta_2} + \beta_1 V_{\alpha_1, \beta_1+1, \alpha_2, \beta_2}}{\alpha_1 + \beta_1}.$$

Thanks to the tractability of the one-armed case, where one of the two Beta distributions collapses to a point mass, $V'$ satisfies two boundary conditions. This suggests a natural approximation scheme for implementing the $\mathcal{R}^2$-optimal policy: (i) impose the two boundary conditions on $\{(\alpha_1, \beta_1, \alpha_2, \beta_2) : \alpha_1 + \beta_1 = \bar{M} \text{ or } \alpha_2 + \beta_2 = \bar{M}\}$, where $\bar{M}$ is finite; (ii) propagate the values of $V'$ inward via the Bellman equation (2); and (iii) extract a policy from $V'$.

In Figure 3, we compare Thompson Sampling with the $\mathcal{R}^2$-optimal policy (with different values of $\bar{M}$) in the two-armed case. For each value of $\bar{M}$, let the corresponding policy play $\bar{M}/2$ rounds. The left panel of Figure 2 shows that the regret curves corresponding to different values of $\bar{M}$ are closely aligned, indicating that the $\bar{M}$-truncation already captures the behavior of the $\mathcal{R}^2$-optimal policy over the first $\bar{M}/2$ rounds. After 20 rounds, the $\mathcal{R}^2$-optimal policy achieves a 30% reduction in cumulative regret relative to Thompson Sampling. Again, the reason behind the gap can be understood via regularizer comparison. In the right panel of Figure 3, we compare the two regularizers along the prior sequence $\pi_0(k) = \text{Beta}(5,4) \times \text{Beta}(k,k)$, with $k = 1, \ldots, 7$, to illustrate how the two policies re-

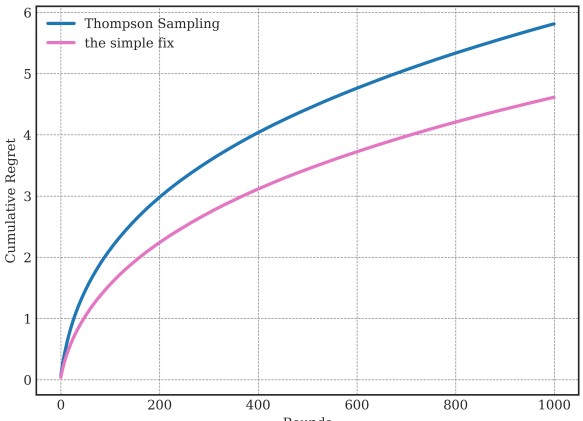 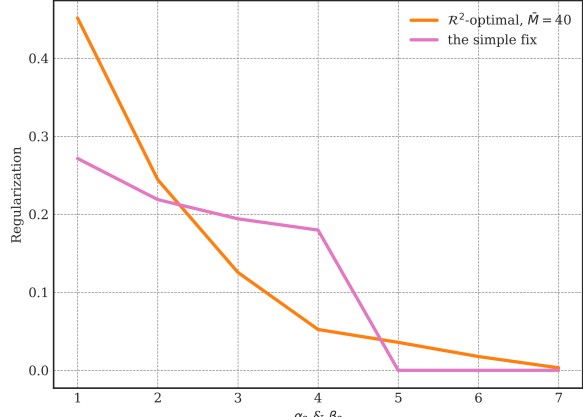

*Figure 4.* Left: Compare Thompson Sampling and the simple fix in terms of cumulative regret $\mathcal{R}_T(Q^{\mathrm{TS}}; \pi_0)$ vs. $\mathcal{R}_T(Q^{\mathrm{FIX}}; \pi_0)$ where $\pi_0 = \mathrm{Beta}(5, 4) \times \mathrm{Beta}(500, 500)$ (20K trials). Right: Comparing the simple fix and the $\mathcal{R}^2$-optimal policy in terms of regularizers $\nu^{\mathrm{FIX}}(\mathrm{Beta}(5, 4) \times \mathrm{Beta}(k, k))$ vs. $\nu^{\mathrm{R2}}(\mathrm{Beta}(5, 4) \times \mathrm{Beta}(k, k))$ where $k = 1, \ldots, 7$.

spond as the under-performing arm gradually becomes over-explored. For arm 1, the mean of $\mathrm{Beta}(5, 4)$ exceeds $1/2$. For arm 2, the mean of $\mathrm{Beta}(k, k)$ remains $1/2$ (under-performing), while the distribution becomes increasingly concentrated (over-explored) as $k$ increases. In particular, when $\mathrm{Beta}(5, 4)$ vs. $\mathrm{Beta}(7, 7)$, arm 1 receives fewer pulls ($5 + 4 < 7 + 7$) but has a higher posterior mean reward $5/9$, making it attractive from both exploration and exploitation perspectives. The right panel of Figure 3 shows that the regularizer of the $\mathcal{R}^2$-optimal policy decays quickly, thereby encouraging fully greedy behavior (i.e., pulling arm 1 with probability one). In contrast, the regularizer of Thompson Sampling does not decay fast enough to temporarily stop pulling arm 2, which is under-performing and over-explored. Again, this conservativeness is because the uncertainty measure $\nu^{\mathrm{TS}}(\pi_t)$ does not fully capture the tension between exploration and exploitation quantified by $\nu^{\mathrm{R2}}(\pi_t)$. As the under-performing arm gradually becomes over-explored, the tension vanishes while the uncertainty (now primarily contributed by the other arm) remains.

## 6. A Simple Fix

Recall that, through the online optimization lens, Thompson Sampling regularizes greediness according to uncertainty, whereas the $\mathcal{R}^2$-optimal policy does so according to tension. The comparative analysis so far shows that the remaining uncertainty (about which arm is better) is *not* a sufficient proxy for the current tension (between exploration and exploitation), which in turn leads to a substantial performance gap between the two policies. Motivated by this insight, we propose a simple fix that makes the uncertainty measure more tension-aware:

$$\nu^{\mathrm{FIX}}(\pi_t) = (1 - s(\pi_t))\nu^{\mathrm{TS}}(\pi_t),$$

where $s(\pi_t)$ is a switch that shuts down regularization when there is no tension. The shutdown criterion is given by

$$s(\pi_t) = \sum_{k=1}^{2} \mathbb{I}(\mathbb{E}_{\pi_t}\theta_k > \mathbb{E}_{\pi_t}\theta_{3-k}) \, \mathbb{I}(\mathcal{V}_k(\pi_t) > \mathcal{V}_{3-k}(\pi_t)).$$

where $\mathcal{V}_k(\pi_t) = \mathrm{Var}_{\pi_t}\mathbb{E}_{\pi_t}(\theta_k|\Lambda)$ is the (variance-based) information gain from pulling arm $k$ (Russo & Van Roy, 2014a). When $s(\pi_t) = 1$, the arm giving more reward also gives more information (hence there is no tension between exploration and exploitation), rendering regularization unnecessary. It is straightforward to show that the resulting policy $Q^{\mathrm{FIX}}$ remains $\mathcal{R}^2$-finite.

**Proposition 6.1** ($\mathcal{R}^2$-finiteness of $Q^{\mathrm{FIX}}$). *In the setting of Proposition 3.1, the simple fix satisfies $\mathcal{R}^2(Q^{\mathrm{FIX}}; \pi_0) < \infty$.*

On the one hand, the right panel of Figure 4 shows that the shutdown mechanism brings the regularizer of Thompson Sampling into closer alignment with the Bellman-optimal benchmark. On the other hand, the left panel of Figure 4 shows that the shutdown mechanism indeed reduces the cumulative regret of Thompson Sampling when the under-performing arm is over-explored. Altogether, this example confirms the appeal of a principled framework with a well-defined benchmark, which serves as a compass for improvement. The closer a regularizer is to the benchmark, the better the performance of its corresponding policy.

## 7. A Principled Fix

Given the Bellman equation (2), we can perform policy evaluation, improvement, and iteration. With $V^{\mathrm{TS}}(\cdot) = \mathcal{R}^2(Q^{\mathrm{TS}}; \cdot)$, the one-step improved Thompson Sampling is

$$q_t^{\mathrm{TS}'} = \underset{q_t}{\mathrm{argmin}}[r^2(q_t; \pi_t) + q_t \cdot \mathbb{E}_{\pi_t}[V^{\mathrm{TS}}(\pi_{t+1})|A_t = \cdot]].$$

For a detailed discussion of this principled fix, please see (Qu et al., 2025), the full version of the current paper.

## 8. Conclusion

Thompson Sampling can be viewed as an online optimization algorithm for squared regret minimization, offering a new perspective for understanding and improving this famous heuristic bandit algorithm. By comparing it against the Bellman-optimal benchmark, its issues can be identified and addressed in a principled manner.

## Impact Statement

This paper presents work whose goal is to advance the field of multi-armed bandits. There are many potential societal consequences of our work, none of which we feel must be specifically highlighted here.

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

# A. Proofs

*Proof of Proposition 3.1.* By the information-ratio bound of (Russo & Van Roy, 2016), under the stated $\sigma$-sub-Gaussian assumption, we have

$$r^2(q_t^{\text{TS}}; \pi_t) \leq 2K\sigma^2 \cdot \mathbb{E}_{\pi_t}\left[D_{\text{KL}}(q_{t+1}^{\text{TS}} \| q_t^{\text{TS}})\right],$$

where $K$ is the number of arms, and $D_{\text{KL}}(\cdot\|\cdot)$ is the KL divergence. Since

$$\mathbb{E}_{\pi_t} q_{t+1}^{\text{TS}} = \mathbb{E}_{\pi_t} P_{\pi_t}\left(\text{argmax}\{\theta_1, ...\theta_K\} = \cdot | \pi_{t+1}\right) = P_{\pi_t}\left(\text{argmax}\{\theta_1, ...\theta_K\} = \cdot\right) = q_t^{\text{TS}},$$

we have

$$\mathbb{E}_{\pi_t}\left[D_{\text{KL}}(q_{t+1}^{\text{TS}} \| q_t^{\text{TS}})\right] = \mathbb{E}_{\pi_t}\left[\sum_{k=1}^{K} q_{k,t+1}^{\text{TS}} \log \frac{q_{k,t+1}^{\text{TS}}}{q_{k,t}^{\text{TS}}}\right]$$

$$= \mathbb{E}_{\pi_t}\left[\sum_{k=1}^{K} q_{k,t+1}^{\text{TS}} \log q_{k,t+1}^{\text{TS}}\right] - \sum_{k=1}^{K} q_{k,t}^{\text{TS}} \log q_{k,t}^{\text{TS}}$$

$$= H(q_t^{\text{TS}}) - \mathbb{E}_{\pi_t} H(q_{t+1}^{\text{TS}}),$$

where $H(\cdot)$ is the entropy. Therefore, we have

$$\mathcal{R}^2(Q^{\text{TS}}; \pi_0) = \mathbb{E}_{\pi_0}\left[\sum_{t=0}^{\infty} r^2(q_t^{\text{TS}}; \pi_t)\right]$$

$$\leq 2K\sigma^2 \cdot \mathbb{E}_{\pi_0}\left[\sum_{t=0}^{\infty}\left[H(q_t^{\text{TS}}) - \mathbb{E}_{\pi_t} H(q_{t+1}^{\text{TS}})\right]\right]$$

$$= 2K\sigma^2 \cdot \sum_{t=0}^{\infty}\left[\mathbb{E}_{\pi_0} H(q_t^{\text{TS}}) - \mathbb{E}_{\pi_0} H(q_{t+1}^{\text{TS}})\right]$$

$$\leq 2K\sigma^2 \cdot H(q_0^{\text{TS}})$$

$$< \infty.$$

$\square$

*Proof of Theorem 4.2.* Recall that $\Delta = \theta_1 - \theta_2$, $\Lambda = \text{sign}(\theta_1 - \theta_2)$, and

$$\frac{\text{Cov}_{\pi_t}(\Delta, \Lambda)}{2} = \mathbb{E}_{\pi_t} \Delta I(\Delta > 0) - P_{\pi_t}(\Delta > 0)\mathbb{E}_{\pi_t}\Delta$$

$$= P_{\pi_t}(\Delta \leq 0)\mathbb{E}_{\pi_t}\Delta I(\Delta > 0) + P_{\pi_t}(\Delta > 0)\mathbb{E}_{\pi_t}\Delta I(\Delta > 0)$$

$$- P_{\pi_t}(\Delta > 0)\mathbb{E}_{\pi_t}\Delta I(\Delta > 0) - P_{\pi_t}(\Delta > 0)\mathbb{E}_{\pi_t}\Delta I(\Delta \leq 0)$$

$$= P_{\pi_t}(\Delta \leq 0)\mathbb{E}_{\pi_t}\Delta I(\Delta > 0) - P_{\pi_t}(\Delta > 0)\mathbb{E}_{\pi_t}\Delta I(\Delta \leq 0).$$

By differentiation, the minimizer of the online objective is

$$\mathbb{E}_{\pi_t} \max(\theta_1, \theta_2) - \frac{\text{Cov}_{\pi_t}(\Delta, \Lambda)}{2}$$

$$= P_{\pi_t}(\Delta \leq 0)(\mathbb{E}_{\pi_t}\theta_2 + \mathbb{E}_{\pi_t}\Delta I(\Delta > 0)) + P_{\pi_t}(\Delta > 0)(\mathbb{E}_{\pi_t}\theta_1 - \mathbb{E}_{\pi_t}\Delta I(\Delta \leq 0))$$

$$- P_{\pi_t}(\Delta \leq 0)\mathbb{E}_{\pi_t}\Delta I(\Delta > 0) + P_{\pi_t}(\Delta > 0)\mathbb{E}_{\pi_t}\Delta I(\Delta \leq 0)$$

$$= P_{\pi_t}(\Delta > 0)\mathbb{E}_{\pi_t}\theta_1 + P_{\pi_t}(\Delta \leq 0)\mathbb{E}_{\pi_t}\theta_2,$$

which is the expected next-round reward of Thompson Sampling.

$\square$

*Proof of Proposition 4.3.* For $\lambda \neq 1$, the unconstrained minimizer of the online objective is

$$\bar{x}_t^{\lambda} = \mathbb{E}_{\pi_t} \max(\theta_1, \theta_2) - \frac{\lambda \text{Cov}_{\pi_t}(\Delta, \Lambda)}{2}$$

$$= \lambda\left(\mathbb{E}_{\pi_t} \max(\theta_1, \theta_2) - \frac{\text{Cov}_{\pi_t}(\Delta, \Lambda)}{2}\right) + (1 - \lambda)\mathbb{E}_{\pi_t} \max(\theta_1, \theta_2)$$

$$= \lambda\left(P_{\pi_t}(\Delta > 0)\mathbb{E}_{\pi_t}\theta_1 + P_{\pi_t}(\Delta \leq 0)\mathbb{E}_{\pi_t}\theta_2\right) + (1 - \lambda)\mathbb{E}_{\pi_t} \max(\theta_1, \theta_2).$$

The constrained minimizer $x_t^\lambda$ is obtained by clipping $\bar{x}_t^\lambda$ to be between $\mathbb{E}_{\pi_t}\theta_1$ and $\mathbb{E}_{\pi_t}\theta_2$. When $\pi_t = \delta_{1-\lambda} \times N(0, \sigma^2)$, we have

$$\mathbb{E}_{\pi_t}\max(\theta_1, \theta_2) = \mathbb{E}\max(1 - \lambda, N(0, \sigma^2)) = \sigma\mathbb{E}\max((1 - \lambda)/\sigma, N(0, 1)) \to \infty,$$

as $\sigma \to \infty$. When $\sigma$ is large enough, we have

$$\lambda < 1 \Rightarrow \bar{x}_t^\lambda > \mathbb{E}_{\pi_t}\theta_1 > \mathbb{E}_{\pi_t}\theta_2 \Rightarrow x_t^\lambda = \mathbb{E}_{\pi_t}\theta_1,$$
$$\lambda > 1 \Rightarrow \bar{x}_t^\lambda < \mathbb{E}_{\pi_t}\theta_1 < \mathbb{E}_{\pi_t}\theta_2 \Rightarrow x_t^\lambda = \mathbb{E}_{\pi_t}\theta_1.$$

In either case, arm 1 is pulled with probability one, but pulling the known arm 1 produces no posterior update. Consequently, the policy fully commits to arm 1 while arm 2 still has a chance of being better. □

*Proof of Proposition 4.4.* When $\mathrm{Var}_{\pi_t}\Lambda > 0$, we have

$$\frac{\mathrm{Cov}_{\pi_t}(\Delta, \Lambda)}{\mathrm{Var}_{\pi_t}\Lambda} = \frac{2P_{\pi_t}(\Delta \le 0)\mathbb{E}_{\pi_t}\Delta I(\Delta > 0)}{4P_{\pi_t}(\Delta > 0)P_{\pi_t}(\Delta \le 0)} - \frac{2P_{\pi_t}(\Delta > 0)\mathbb{E}_{\pi_t}\Delta I(\Delta \le 0)}{4P_{\pi_t}(\Delta > 0)P_{\pi_t}(\Delta \le 0)} = \frac{\mathbb{E}_{\pi_t}[\Delta|\Delta > 0] + \mathbb{E}_{\pi_t}[-\Delta|\Delta \le 0]}{2}.$$

□

*Proof of Proposition 5.1.* When $\theta_2 \equiv 0$, the Bellman equation (2) becomes

$$0 = \min_{q_t}\left[\left(\mathbb{E}_{\pi_t}[(\theta_1)_+] - q_{1,t}\mathbb{E}_{\pi_t}\theta_1\right)^2 + q_{1,t}\left(\mathbb{E}_{\pi_t}[V(\pi_{t+1})|A_t = 1] - V(\pi_t)\right) + q_{2,t}\left(\mathbb{E}_{\pi_t}[V(\pi_{t+1})|A_t = 2] - V(\pi_t)\right)\right]$$
$$= \min_{q_{1,t}}\left[\left(\mathbb{E}_{\pi_t}[(\theta_1)_+] - q_{1,t}\mathbb{E}_{\pi_t}\theta_1\right)^2 - q_{1,t}\left(V(\pi_t) - \mathbb{E}_{\pi_t}[V(\pi_{t+1})|A_t = 1]\right)\right],$$

where $\mathbb{E}_{\pi_t}[V(\pi_{t+1})|A_t = 2] - V(\pi_t) = 0$ as pulling the known arm 2 produces no posterior update. Note that the above minimization is equivalent to

$$V(\pi_t) - \mathbb{E}_{\pi_t}[V(\pi_{t+1})|A_t = 1] = \min_{q_{1,t}}\left[\frac{\left(\mathbb{E}_{\pi_t}[(\theta_1)_+] - q_{1,t}\mathbb{E}_{\pi_t}\theta_1\right)^2}{q_{1,t}}\right].$$

The minimizer of

$$\frac{\left(\mathbb{E}_{\pi_t}[(\theta_1)_+] - q_{1,t}\mathbb{E}_{\pi_t}\theta_1\right)^2}{q_{1,t}} = \frac{\left(\mathbb{E}_{\pi_t}[(\theta_1)_+]\right)^2}{q_{1,t}} + q_{1,t}\left(\mathbb{E}_{\pi_t}\theta_1\right)^2 - 2\mathbb{E}_{\pi_t}[(\theta_1)_+]\mathbb{E}_{\pi_t}\theta_1$$

in $[0, 1]$ is clearly

$$q_{1,t}^{\mathrm{R2}} = \min\left(\frac{\mathbb{E}_{\pi_t}[(\theta_1)_+]}{|\mathbb{E}_{\pi_t}\theta_1|}, 1\right).$$

□

*Proof of Proposition 5.2.* When $\theta_2 \equiv 0$ and $\theta_1 \sim N(\mu_t, \sigma_t^2)$ under $\pi_t$, we have

$$\begin{aligned}
q_{1,t}^{\mathrm{R2}} = 1 &\Leftrightarrow \mathbb{E}_{\pi_t}[(\theta_1)_+] \ge |\mathbb{E}_{\pi_t}\theta_1| \\
&\Leftrightarrow \mathbb{E}_{\pi_t}[(\theta_1)_+] \ge -\mathbb{E}_{\pi_t}\theta_1 \\
&\Leftrightarrow \mathbb{E}_{\pi_t}[(\theta_1)_+] + \mathbb{E}_{\pi_t}\theta_1 \ge 0 \\
&\Leftrightarrow \mu_t\Phi\left(\frac{\mu_t}{\sigma_t}\right) + \sigma_t\phi\left(\frac{\mu_t}{\sigma_t}\right) + \mu_t \ge 0 \\
&\Leftrightarrow \frac{\mu_t}{\sigma_t}\Phi\left(\frac{\mu_t}{\sigma_t}\right) + \phi\left(\frac{\mu_t}{\sigma_t}\right) + \frac{\mu_t}{\sigma_t} \ge 0 \\
&\Leftrightarrow \frac{\mu_t}{\sigma_t} \ge \bar{x},
\end{aligned}$$

where $\bar{x} \approx -0.276$ is the unique root of the increasing function $x\Phi(x) + \phi(x) + x$. □

*Proof of Proposition 6.1.* It suffices to show that the information ratio of $Q^{\mathrm{FIX}}$ is bounded by that of $Q^{\mathrm{TS}}$. When the two policies act differently, we have $s(\pi_t) = 1$, which corresponds to two cases. By symmetry, we consider the case where $\mathbb{E}_{\pi_t}\theta_1 > \mathbb{E}_{\pi_t}\theta_2$ and $\mathcal{V}_1(\pi_t) > \mathcal{V}_2(\pi_t)$. In this case, we have $q_{1,t}^{\mathrm{FIX}} = 1$ and

$$\frac{(\mathbb{E}_{\pi_t}\max(\theta_1, \theta_2) - \mathbb{E}_{\pi_t}\theta_1)^2}{\mathcal{V}_1(\pi_t)} \leq \frac{(\mathbb{E}_{\pi_t}\max(\theta_1, \theta_2) - (q_{1,t}^{\mathrm{TS}}\mathbb{E}_{\pi_t}\theta_1 + q_{2,t}^{\mathrm{TS}}\mathbb{E}_{\pi_t}\theta_2))^2}{q_{1,t}^{\mathrm{TS}}\mathcal{V}_1(\pi_t) + q_{2,t}^{\mathrm{TS}}\mathcal{V}_2(\pi_t)},$$

as the left-hand side has a smaller numerator and a larger denominator than the right-hand side. $\square$

# B. Implementation Details

In this appendix, we describe in detail how to approximately implement the $\mathcal{R}^2$-optimal policy for two-armed Bernoulli bandits[1]. In the Beta–Bernoulli setting, the reward distribution is Bernoulli, while both the prior and posterior are Beta, so the belief space is parametrized by two pairs of positive integers $\{\mathrm{Beta}(\alpha_1, \beta_1) \times \mathrm{Beta}(\alpha_2, \beta_2) : \alpha_1, \beta_1, \alpha_2, \beta_2 \geq 1\}$. Let $V_{\alpha_1,\beta_1,\alpha_2,\beta_2} = V(\mathrm{Beta}(\alpha_1, \beta_1) \times \mathrm{Beta}(\alpha_2, \beta_2))$ be the solution to the Bellman equation

$$
\begin{aligned}
V_{\alpha_1,\beta_1,\alpha_2,\beta_2} = \min_{p,q} \Big[ & (E_{\alpha_1,\beta_1,\alpha_2,\beta_2} - (pE_{\alpha_1,\beta_1} + qE_{\alpha_2,\beta_2}))^2 \\
& + p(E_{\alpha_1,\beta_1}V_{\alpha_1',\beta_1,\alpha_2,\beta_2} + \bar{E}_{\alpha_1,\beta_1}V_{\alpha_1,\beta_1',\alpha_2,\beta_2}) \\
& + q(E_{\alpha_2,\beta_2}V_{\alpha_1,\beta_1,\alpha_2',\beta_2} + \bar{E}_{\alpha_2,\beta_2}V_{\alpha_1,\beta_1,\alpha_2,\beta_2'}) \Big],
\end{aligned}
$$

where $p + q = 1$, $\alpha_1' = \alpha_1 + 1$, $E_{\alpha_1,\beta_1} = \alpha_1/(\alpha_1 + \beta_1)$, $\bar{E}_{\alpha_1,\beta_1} = 1 - E_{\alpha_1,\beta_1}$, and

$$E_{\alpha_1,\beta_1,\alpha_2,\beta_2} = \mathbb{E}\max(\mathrm{Beta}(\alpha_1, \beta_1), \mathrm{Beta}(\alpha_2, \beta_2)).$$

**Benefit function.** The "benefit" of pulling arm 1 is

$$V_{\alpha_1,\beta_1,\alpha_2,\beta_2}' = V_{\alpha_1,\beta_1,\alpha_2,\beta_2} - E_{\alpha_1,\beta_1}V_{\alpha_1',\beta_1,\alpha_2,\beta_2} - \bar{E}_{\alpha_1,\beta_1}V_{\alpha_1,\beta_1',\alpha_2,\beta_2}.$$

As $V$ is symmetric (i.e., $V_{\alpha_1,\beta_1,\alpha_2,\beta_2} = V_{\alpha_2,\beta_2,\alpha_1,\beta_1}$), the benefit of pulling arm 2 is simply $V_{\alpha_2,\beta_2,\alpha_1,\beta_1}'$. Note that the Bellman equation for $V$ can be rewritten in terms of $V'$

$$V_{\alpha_2,\beta_2,\alpha_1,\beta_1}' = \min_{p,q} \Big[ (E_{\alpha_1,\beta_1,\alpha_2,\beta_2} - (pE_{\alpha_1,\beta_1} + qE_{\alpha_2,\beta_2}))^2 - p(V_{\alpha_1,\beta_1,\alpha_2,\beta_2}' - V_{\alpha_2,\beta_2,\alpha_1,\beta_1}') \Big],$$

so the $\mathcal{R}^2$-optimal policy can be characterized by $V'$. Therefore, our goal becomes to compute $V'$.

**Backward recursion.** Note that $V'$ satisfies

$$V_{\alpha_1,\beta_1,\alpha_2,\beta_2}' - E_{\alpha_2,\beta_2}V_{\alpha_1,\beta_1,\alpha_2',\beta_2}' - \bar{E}_{\alpha_2,\beta_2}V_{\alpha_1,\beta_1,\alpha_2,\beta_2'}' = V_{\alpha_2,\beta_2,\alpha_1,\beta_1}' - E_{\alpha_1,\beta_1}V_{\alpha_2,\beta_2,\alpha_1',\beta_1}' - \bar{E}_{\alpha_1,\beta_1}V_{\alpha_2,\beta_2,\alpha_1,\beta_1'}',$$

as both sides equal to

$$
\begin{aligned}
& V_{\alpha_1,\beta_1,\alpha_2,\beta_2} - E_{\alpha_1,\beta_1}V_{\alpha_1',\beta_1,\alpha_2,\beta_2} - \bar{E}_{\alpha_1,\beta_1}V_{\alpha_1,\beta_1',\alpha_2,\beta_2} - E_{\alpha_2,\beta_2}V_{\alpha_1,\beta_1,\alpha_2',\beta_2} - \bar{E}_{\alpha_2,\beta_2}V_{\alpha_1,\beta_1,\alpha_2,\beta_2'} \\
& + E_{\alpha_1,\beta_1}E_{\alpha_2,\beta_2}V_{\alpha_1',\beta_1,\alpha_2',\beta_2} + \bar{E}_{\alpha_1,\beta_1}\bar{E}_{\alpha_2,\beta_2}V_{\alpha_1,\beta_1',\alpha_2,\beta_2'} + \bar{E}_{\alpha_1,\beta_1}E_{\alpha_2,\beta_2}V_{\alpha_1,\beta_1',\alpha_2',\beta_2} + E_{\alpha_1,\beta_1}\bar{E}_{\alpha_2,\beta_2}V_{\alpha_1',\beta_1,\alpha_2,\beta_2'},
\end{aligned}
$$

which remains unchanged when subscripts 1 and 2 are swapped. This identity allows us to compute the difference $V_{\alpha_1,\beta_1,\alpha_2,\beta_2}' - V_{\alpha_2,\beta_2,\alpha_1,\beta_1}'$ from $V_{\alpha_1,\beta_1,\alpha_2',\beta_2}'$, $V_{\alpha_1,\beta_1,\alpha_2,\beta_2'}'$, $V_{\alpha_2,\beta_2,\alpha_1',\beta_1}'$, $V_{\alpha_2,\beta_2,\alpha_1,\beta_1'}'$. Then we can compute $V_{\alpha_1,\beta_1,\alpha_2,\beta_2}'$ and $V_{\alpha_2,\beta_2,\alpha_1,\beta_1}'$ by plugging the difference into the Bellman equation for $V'$.

**Two boundary conditions.** When $\mathrm{Beta}(\alpha_1, \beta_1)$ collapses to a point mass ($\alpha_1 + \beta_1 = \infty$), we have $V_{\alpha_1,\beta_1,\alpha_2,\beta_2}' = 0$, as pulling the known arm 1 brings no further benefit. When $\mathrm{Beta}(\alpha_2, \beta_2)$ collapses to a point mass ($\alpha_2 + \beta_2 = \infty$), we similarly have $V_{\alpha_2,\beta_2,\alpha_1,\beta_1}' = 0$, and the Bellman equation for $V'$ reduces to

$$V_{\alpha_1,\beta_1,\alpha_2,\beta_2}' = \min_{p,q} \left[ \frac{(E_{\alpha_1,\beta_1,\alpha_2,\beta_2} - (pE_{\alpha_1,\beta_1} + qE_{\alpha_2,\beta_2}))^2}{p} \right].$$

**Finite truncation.** Given the backward recursion and the two boundary conditions, a natural approximation scheme for computing $V'$ is as follows: (i) impose the two boundary conditions on $\{(\alpha_1, \beta_1, \alpha_2, \beta_2) : \alpha_1 + \beta_1 = \bar{M} \text{ or } \alpha_2 + \beta_2 = \bar{M}\}$, where $\bar{M}$ is finite; and (ii) propagate the values of $V'$ inward via the backward recursion.

---

[1]Code is available at https://github.com/quyanlin/what-does-thompson-sampling-optimize.

