# OpenReview forum: "What Does Thompson Sampling Optimize?"
_ICML.cc/2026/Conference — ICML 2026 regular_

### Official Review · Reviewer_9tyT · 2026-02-25

**Soundness:** 4
**Presentation:** 4
**Significance:** 4
**Originality:** 4
**Overall Recommendation:** 6
**Confidence:** 5

**Summary:**

Thompson sampling (TS) is generally discussed as heuristic method for tackling multi-armed bandit problems. This paper puts TS on a more principled footing, describing it as an optimization problem. The optimization is presented as mirroring the structure of the Bellman-optimal problem, then empirical comparisons are made. Finally, the authors present a "simple fix" that makes TS behave more like the Bellman version and improves its performance. This serves to demonstrate their point that a principled, variational construction of TS opens the door to systematic improvement of TS.

**Compliance With Llm Reviewing Policy:**

Affirmed.

**Final Justification:**

All concerns addressed.

**Key Questions For Authors:**

None

**Limitations:**

Reproducibility of empirical results: Point to a github repo.

**Strengths And Weaknesses:**

This is an interesting and important question. The motivation is clear, as is the development. The comparison between the Bellman optimal algorithm and TS offers insight into suboptimal behavior of TS. The "simple fix" is both.

The use of regret^2 is central and, while clearly and correctly represented in the paper, readers may be so primed to expect discussion of regret (not regret^2) that this distinction and the motivation for it might benefit from extended discussion.

The empirical studies, while sufficient to make the points, seem somewhat scant. At least, they would benefit from making the code available for its value in reproducibility and communication of technical details.

This paper is original and interesting and I enjoyed reading it. I think this paper will hold a significant place in Thompson sampling literature.

---

> ### Author Rebuttal · Authors · 2026-03-31
>
> Thank you for your valuable feedback and positive review about our paper. We agree that the leap from regret to squared regret deserves more discussion. Why do we want to stationarize the long-term regret minimization problem (finite horizon)?  It can be stationarized by discounting, why do we have to develop a different one (squared regret)? Please see our response to Reviewer kH1z (question 1) where we answer these questions. Why squared? Why not consider $p$-th power regret for $p\neq2$? Please see our response to Reviewer BHNu (question 2) where we answer these questions. We thank the reviewer for raising this point, and we will add these discussions to the camera-ready version.
>
> We are glad that the empirical studies are sufficient to make the points, as our stylized experiments are intended to visualize key insights. The code is included in the supplementary material, with four Jupyter notebooks corresponding to Figures 1-4. We are also happy to include a GitHub link in the final version for ease of access.

---

> > ### Author Rebuttal · Reviewer_9tyT · 2026-04-01
> >
> > Resolved.

---

### Official Review · Reviewer_kH1z · 2026-03-08

**Soundness:** 4
**Presentation:** 4
**Significance:** 4
**Originality:** 4
**Overall Recommendation:** 5
**Confidence:** 4

**Summary:**

The paper reframes Thompson Sampling as an online optimisation problem and defining the Bellman function for that problem. This definition is then used to better understand how Thompson Sampling regularises immediate reward vs exploration. This is compared against a Bellman-optimal benchmark, and the paper then proposes a modification to the regulariser based on this understanding to improve performance. This is shown on small problem instances empirically.

**Compliance With Llm Reviewing Policy:**

Affirmed.

**Final Justification:**

After the rebuttal and discussion, I remain firmly in favour of acceptance, with increased confidence.

The rebuttal successfully addresses my main concern about the “simple fix” by introducing a principled policy-improvement interpretation grounded in the Bellman framework. This strengthens the paper by showing that the algorithmic contribution follows naturally from the proposed squared-regret formulation, rather than being ad-hoc.

The core contribution, a reinterpretation of Thompson Sampling via a Bellman perspective, is in my view both original and insightful. The analysis of the induced regularisation and its connection to exploration is very interesting, and the additional clarification on squared regret and stationarisation further supports the framework.

Overall, this is a strong and conceptually impactful paper, and I am happy to support acceptance.

**Key Questions For Authors:**

- Can you provide further intuition for why squared regret, rather than standard regret or discounted regret, is the “right” objective that Thompson Sampling is implicitly optimizing? Is there a deeper principle that links to squared regret, or is this primarily a mathematical device that makes the Bellman formulation tractable?
- Have you evaluated whether the proposed tension-aware regularizer improves performance beyond the specific illustrative scenarios?

**Limitations:**

Yes, these are adequately discussed

**Strengths And Weaknesses:**

- Strengths:
- A strong and original conceptual contribution reinterpreting Thompson Sampling providing a new way of looking at Thompson Sampling with a clear structural comparison to dynamic programming.
- Interesting identification of the TS regulariser that deepens understanding of why TS explores.
- Interesting benchmarking against the Bellman optimal solution.
- Well written and clear text with solid and coherent technical depth.
- Weaknesses:
- The algorithmic ‘simple fix’ is interesting, but discussed more as a ad-hoc heuristic than with as much depth and soundness as the rest of the paper. In general, it would be interesting to have a more practical and wider view (besides the smaller problem instances here) on how to use the insights.

---

> ### Author Rebuttal · Authors · 2026-03-31
>
> Thank you for your valuable feedback and positive review about our paper. To address the concerns raised, we report several new findings that further highlight the benefit of our online optimization perspective on Thompson Sampling and, more broadly, on the Bayesian bandit problem. In particular, under our squared regret formulation, Bellman’s *policy improvement* can now be applied to Thompson Sampling.
>
> We agree that the “simple fix” looks more ad-hoc than the rest of the paper, so we develop a principled fix: policy improvement. (Thanks to our squared regret formulation, this reinforcement learning terminology begins to make sense in the bandit setting.) This illustrates how our new perspective leads to algorithmic improvement in a Bellman-principled manner. Here is a description.
>
> Let $V^\mathrm{TS}(\cdot)=\mathcal{R}^2(Q^\mathrm{TS};\cdot)$ be the squared-regret value function of Thompson Sampling. Plugging it into the right-hand side of the Bellman equation (2) produces the one-step improved Thompson Sampling
> $$
> q_t^{\mathrm{TS}’}=\underset{q_t}{\mathrm{argmin}}\left[r^2(q_t;\pi_t)+q_t\cdot E_{\pi_t}\left[V^\mathrm{TS}(\pi_{t+1})|A_t=\cdot\right]\right],
> $$
> which achieves lower squared regret than Thompson Sampling. If we apply more policy-improvement steps, we have
> $$
> \mathcal{R}^2(Q^\mathrm{TS};\cdot)\geq\mathcal{R}^2(Q^{\mathrm{TS}'};\cdot)\geq\mathcal{R}^2(Q^{\mathrm{TS}''};\cdot)\geq...\geq\mathcal{R}^2(Q^\mathrm{R2};\cdot),
> $$
> which successively reduces the leading constant in the regret bound (1). It turns out that the first step already improves Thompson Sampling significantly.
>
> In this figure (https://imgur.com/a/KYlv9xf), we compare Thompson Sampling before and after a single policy-improvement step. The striking similarity between this figure ($Q^\mathrm{TS}$ vs. $Q^{\mathrm{TS}’}$) and Figure 3 in the paper ($Q^\mathrm{TS}$ vs. $Q^\mathrm{R2}$) shows that this single step already brings Thompson Sampling quite close to the Bellman-optimal benchmark. The left panel shows that it closes a large proportion (about 90%) of the performance gap between Thompson and Bellman. The right panel shows it transforms the regularizer of Thompson Sampling from an uncertainty measure to a tension measure. (For this drastic structural change, we have some new insights in the $K$-arm case ($K>2$); please see our response to Reviewer BHNu (question 1).) Altogether, this example illustrates the power of first principles.
>
> Now we do have a more practical and wider view on how to use the insights: for any $\mathcal{R}^2$-finite policy handling any number of arms, multi-armed bandit policy evaluation, improvement, and iteration at scale (e.g., storing $V^\mathrm{TS}$, and hence $Q^{\mathrm{TS}’}$, in a neural network) is a natural direction for future research.
>
> **Answer to question 1:** Can you provide further intuition for why squared regret, rather than standard regret or discounted regret, is the “right” objective that Thompson Sampling is implicitly optimizing? Is there a deeper principle that links to squared regret, or is this primarily a mathematical device that makes the Bellman formulation tractable?
>
> We refer to the intuition behind our squared regret formulation as *faithful stationarization*, which we next explain. In the bandit literature, the standard goal is to minimize cumulative regret across finite horizons {$\{\mathcal{R}_T(Q;\pi_0):T\geq0\}$}, which is a sequence of non-stationary objectives. In practice, the exact horizon is often unknown or indefinite, making horizon-dependent policies less appealing. More importantly, their time dependence reduces computational tractability and obscures structural insight relative to stationary formulations. These considerations motivate the search for a stationary formulation that preserves the essence of minimizing {$\{\mathcal{R}_T(Q;\pi_0):T\geq0\}$}, which we refer to as faithful stationarization. The discounted reward formulation is not “faithful” to long-term regret minimization as the corresponding optimal policy (Gittins index) suffers linear regret (Rothschild,1974). Therefore, we introduce the squared regret formulation. For the deeper principle that links to squared regret, we believe it lies in the minimax optimality of $O(\sqrt{T})$ cumulative regret; please see our response to Reviewer BHNu (question 2).
>
> **Answer to question 2:** Have you evaluated whether the proposed tension-aware regularizer improves performance beyond the specific illustrative scenarios?
>
> The “simple fix” fixes the issues of Thompson Sampling in a simple (tension-free) case. Since it is not very common to run into the simple case, the performance improvement brought by the shutdown mechanism is not very significant in general. In contrast, the principled fix has significant performance improvement in general, as previously shown.
>
> **References**
>
> Rothschild M (1974) A two-armed bandit theory of market pricing. *Journal of Economic Theory* 9(2):185–202.

---

> > ### Author Rebuttal · Reviewer_kH1z · 2026-04-01
> >
> > Thank you for this detailed response (and the other detailed responses as well), very clear and appreciate the expansion. With this I'm more certain to be in favour of accepting this paper.

---

### Official Review · Reviewer_cfJ3 · 2026-03-08

**Soundness:** 3
**Presentation:** 4
**Significance:** 4
**Originality:** 4
**Overall Recommendation:** 5
**Confidence:** 4

**Summary:**

The authors investigate Thompson Sampling (TS) through an optimisation lens, giving new insights into how TS achieves its exploration-exploitation tradeoff. To produce their analysis, the authors rely on 1) formulating the evolution of the agent’s belief as an MDP and 2) the squared regret as the Value-function of the MDP, which allows them to derive a Bellman equation. The Bellman equation reveals an online optimization problem where the tension between exploration and exploitation is clearly highlighted in a regularizer term. The authors show that TS takes the same optimization form as the squared regret optimal policy, allowing to tweak its regularizer term. Finally, the authors conduct (simple, synthetic) empirical experiments to support their theoretical results and provide further, concrete insights on the underlying mechanisms in TS.

**Compliance With Llm Reviewing Policy:**

Affirmed.

**Key Questions For Authors:**

Q: IDS is brought up a few times in the paper; how does IDS compare to TS after the fix provided in the paper?

**Limitations:**

Yes

**Strengths And Weaknesses:**

The authors convincingly establish (cumulative) squared regret minimization as an appropriate objective to capture bandit performance by showing its relationship with the (cumulative) regret, more specifically that policies with finite cumulative squared regret achieve $O(\sqrt{T})$ cumulative regret. Based on existing literature showing that a Bayesian stochastic bandit can be viewed as a MDP, a Bellman equation is introduced for the squared regret minimization objective. The resulting online optimization problem does involve a regularizer term; the authors provide a clear description of how the quantities in the term define exploration/exploitation behaviours. Analyzing the two-arm case, the authors show that TS admits the same online optimization form as the squared regret optimal policy, with its own regularizer term. The resulting framework sheds light on the impact of regularizer adaptations on the TS behaviour. As a basis of comparison, the authors also apply a similar decomposition to Information-Directed Sampling (IDS), highlighting its own regularizer term, showing that it is closer to the regularizer of the optimal policy, and therefore providing insights on why IDS outperforms TS in practice. Empirical characterization is provided using the one-arm case (two-arm bandits with one arm fully known) and two-arm case. These results contribute to making the provided theoretical results more concrete and visual, enabling further understanding of their implications and validating their soundness.

Some theoretical results have their proofs fully deferred to the appendix; sketches of proofs would have been good for completeness of the main paper. More importantly, Proposition 3.1 does not have a proof at all. While it is stated that this result is a corollary of Russo & Van Roy (2016), it is difficult to validate without diving into the work of Russo and Van Roy. If more space is needed to introduce proof sketches, the authors could consider making Figures 2-4 more compact.

The paper is generally clear and well structured. It highlights how TS has been a very difficult-to-understand approach, despite its generally great performance and is thus well-positioned in the literature. As a notable strength, benefitting from their optimization lens, the paper provides many intuitive examples on how TS behaves in certain scenarios, and highlights how tweaks to TS can improve or deteriorate the performance of TS-like strategies.

TS is still one of the most used bandit approaches; it has been extended to various settings and has inspired exploration-exploitation strategies in MDPs. It remains one of the dominant strategies in practice, almost only challenged by strategies that require more computation (e.g., IDS). Yet, the core exploration-exploitation mechanisms are not well understood. The novel analysis perspective introduced in this paper sheds light on these questions. More generally, it provides a theoretically-grounded framework for understanding decision-making strategies in general, which could prove useful beyond TS (as the authors showed with IDS).  To my knowledge the proposed perspective is novel and so is the provided understanding of the TS exploration-exploitation mechanisms.

---

> ### Author Rebuttal · Authors · 2026-03-31
>
> Thank you for your valuable feedback and positive review about our paper. We agree that Proposition 3.1 deserves a proof in the paper. A concise argument, using the notations introduced in Section 4.6, is as follows. Given the current belief $\pi_t$ and the decision variable (the expected next-round reward) $x_t$, recall that $\bar{r}^2(x_t;\pi_t)$ is the instantaneous squared regret while $\mathcal{I}(x_t;\pi_t)$ is a notion of information gain. For Thompson Sampling, Corollary 1 in Russo & Van Roy (2016) shows that
> $$
> \bar{r}^2(x_t^\mathrm{TS};\pi_t)\leq C\cdot\mathcal{I}(x_t^\mathrm{TS};\pi_t),
> $$
> where $C$ is a universal constant. After summing up and taking expectation, we have
> $$
> \mathcal{R}^2(Q^\mathrm{TS};\pi_0)\leq C\cdot E\left[\sum_{t=0}^\infty\mathcal{I}(x_t^\mathrm{TS};\pi_t)\right],
> $$
> which is finite as the total information gain is bounded by the entropy of the prior (i.e., the initial uncertainty about $\theta$); for details, see the proof of Proposition 1 in Russo & Van Roy (2016). For the other results in our paper, we agree that proof sketches are good for completeness. We have not included them in the current version as we aim to maintain a smooth flow of the main ideas. We thank the reviewer for raising this point, and we will add the proof of Proposition 3.1 to the camera-ready version.
>
> **Answer to the question:** IDS is brought up a few times in the paper; how does IDS compare to TS after the fix provided in the paper?
>
> The “simple fix” fixes the issues of Thompson Sampling in a simple (tension-free) case. Since it is not very common to run into the simple case, the performance improvement brought by the shutdown mechanism is not very significant in general (and less significant than IDS). In fact, instead of the simple fix, we now have a principled fix that brings Thompson Sampling quite close to the Bellman-optimal benchmark; please see our response to Reviewer kH1z (paragraphs 2,3,4,5).

---

> > ### Author Rebuttal · Reviewer_cfJ3 · 2026-03-31
> >
> > I have read the author's response and I am still in favour of accepting this paper.

---

### Official Review · Reviewer_BHNu · 2026-03-13

**Soundness:** 3
**Presentation:** 3
**Significance:** 3
**Originality:** 4
**Overall Recommendation:** 5
**Confidence:** 3

**Summary:**

This paper studies the optimization principle underlying Thompson Sampling (TS). The authors introduce a squared regret objective that aggregates regret across horizons and derive a corresponding Bellman-optimal policy. They show that Thompson Sampling can be expressed as an online optimization problem, where instantaneous regret is regularized by a measure of residual uncertainty. This interpretation provides a new perspective on why TS balances exploration and exploitation.

**Compliance With Llm Reviewing Policy:**

Affirmed.

**Key Questions For Authors:**

1) Does the squared regret objective lead to meaningful insights for other bandit algorithms, not only TS?
2) Are there other regret formulations that might yield similarly useful optimization interpretations?
3) What is the the computational cost of Bellman-optimal benchmark?

**Limitations:**

Yes.

**Strengths And Weaknesses:**

Strengths
1) The paper provides a new theoretical interpretation of TS by recasting it as an online optimization algorithm under a squared regret objective. This perspective is novel and may help explain why TS works despite lacking an explicit mechanism for balancing exploration and exploitation.
2) Under the squared regret formulation, the authors show that TS mimics the structure of the Bellman-optimal policy, where greediness is regularized by a measure of residual uncertainty.
3) The authors derive a closed-form solution in the one-armed case and provide an approximate implementation in the two-armed case, offering useful intuition about exploration behavior and the gap between TS and the Bellman-optimal policy.

Weaknesses
1) The main results are presented under Bayesian regret through the squared regret definition. It would be helpful to discuss whether there are corresponding results or interpretations under frequentist regret.
2) The paper briefly discusses how the new perspective may lead to improvements to TS, but the proposed modification appears somewhat heuristic.
3) The experimental evaluation is limited, mainly consisting of simple synthetic bandit settings.

---

> ### Author Rebuttal · Authors · 2026-03-31
>
> Thank you for your valuable feedback and positive review about our paper. To address the concerns raised, we report several new findings that further highlight the benefit of our online optimization perspective on Thompson Sampling and, more broadly, on the Bayesian bandit problem. In particular, under our squared regret formulation, Bellman’s *policy improvement* can now be applied to Thompson Sampling.
>
> W1. We adopt the Bayesian approach to formulate our optimization perspective mainly because the MDP view renders everything “known”, making (online) optimization possible. In particular, we never observe $\theta$, but its (conditional) distribution is *precisely* tracked by the current belief $\pi_t$, i.e., unknown realization is replaced by known distribution, which is not available in the frequentist setting. For future research, it may be interesting to develop an empirical squared regret formulation, where (Bayesian) posterior distributions are replaced by (frequentist) empirical distributions. We thank the reviewer for raising this point, and we will add the discussion to the camera-ready version.
>
> W2. For the proposed modification (the "simple fix"), we agree that it appears somewhat heuristic, so we develop a principled fix: policy improvement. (Thanks to our squared regret formulation, this reinforcement learning terminology begins to make sense in the bandit setting.) This illustrates how our new perspective leads to algorithmic improvement in a Bellman-principled manner. For a description of the principled fix, please see our response to Reviewer kH1z (paragraphs 3,4,5).
>
> W3. The stylized experiments in this paper are primarily intended to visualize key insights: the benefit of minimizing the regret bound (1) (Figures 2 & 3, left) and the difference between uncertainty and tension (Figures 2 & 3, right). Figure 4 (the simple fix) showcases a tentative step from uncertainty to tension. In contrast, the aforementioned principled fix (policy improvement) is a firm step guided by Bellman’s principle, for which we would like to conduct large-scale experiments in future works.
>
> **Answer to question 1:** Does the squared regret objective lead to meaningful insights for other bandit algorithms, not only TS?
>
> Yes, it does. In the $K$-armed case ($K>2$), any policy of the “$q$” online optimization form (line 204, below (2)), including not only $Q^\mathrm{R2}$ but also $Q^{\mathrm{TS}’}$ (one-step improved Thompson Sampling) and even $Q^\mathrm{IDS}$, assigns positive pulling probability to at most two arms at each round. That is, under our squared regret formulation, Bellman’s principle suggests that mixing at most two arms is a highly desirable structural property, and policy improvement grants Thompson Sampling (which mixes all arms) this property in a single step.
>
> Moreover, somewhat surprisingly, we show that all “$q$” online optimization algorithms are doing *one-dimensional* online optimization regardless of the number of arms. The scalar decision variable is (still) the expected next-round reward $x_t$, from which the full pulling probability vector $q_t$, containing at most two positive entries, can be recovered. The geometric intuition is: $K$ arms correspond to $K$ points (vertices) in the plane, and to represent a point on their lower convex envelope, at most two vertices (arms) are needed, as illustrated in this figure (https://imgur.com/a/zeJkE5W).
>
> **Answer to question 2:** Are there other regret formulations that might yield similarly useful optimization interpretations?
>
> Yes, there are. Similar to squared regret, we can define $p$-th power regret for $p>1$. By replacing Cauchy’s inequality with Holder’s inequality, the regret bound (1) becomes
> $$
> \mathcal{R}_T(Q;\pi_0)\leq T^{1-1/p}\cdot\left[\mathcal{R}^p(Q;\pi_0)\right]^{1/p}.
> $$
> - When $p>2$, the regret bound grows faster than $\sqrt{T}$, which is not sharp. However, the $\mathcal{R}^p$-optimal policy may still be useful for studying the tradeoff between (the growth rates of) the expectation and the variance of the cumulative regret.
> - When $p<2$, the regret bound grows slower than $\sqrt{T}$, which would contradict the minimax optimality of $O(\sqrt{T})$ cumulative regret unless $\mathcal{R}^p(Q;\pi_0)=\infty$ for some $\pi_0$.
> - When $p=2$, the regret bound (1) is sharp, and there exists $Q$ (e.g., $Q^\mathrm{TS}$) such that $\mathcal{R}^2(Q;\pi_0)<\infty$ for all $\pi_0$ (Proposition 3.1). This explains why we focus on squared regret in this paper.
>
> **Answer to question 3:** What is the computational cost of Bellman-optimal benchmark?
>
> To approximately implement the Bellman-optimal benchmark, we solve a "truncated" Bellman equation on a finite grid, so the computational cost grows with the grid size ($O(\bar{M}^4)$ in the two-armed case), which is not scalable. This motivates us to pursue MAB policy evaluation, improvement, and iteration using neural networks in the future, as mentioned in our response to Reviewer kH1z (paragraph 5).

---

> > ### Author Rebuttal · Reviewer_BHNu · 2026-04-04
> >
> > Thank you for the detailed and thoughtful responses. I find the discussion on other regret formulations that might yield similarly useful optimization interpretations particularly interesting. Overall, my main concerns have been adequately addressed and I will keep my score.

---

### Decision · Program_Chairs · 2026-04-30

**Decision:**

Accept (regular)

**Comment:**

The submission provides a fresh perspective on Thompson Sampling (TS), by extracting the mean squared regret as a notion of loss at a time round. Subsequently, it leads to an online optimization formulation for minimizing the sum of mean squared regret, and the perspective allows an improvement on TS and a fresh perspective on IDS. While the submission focuses on the Bayesian 2 arm (with some generalization to $K$ arms) case, which are simple cases, I do appreciate the clear insights it brings. Overall, the submission provides novel insights into Bayesian multi-armed bandits, and I support the acceptance of the paper.

Some thoughts for future works: Can we generalize the framework (and the simple fix to TS) to general multi-armed bandit settings, beyond the $K$-armed setting? While the $K$ armed setting is a good starting point, it will be good to propose some ways forward for general bandit settings in the conclusion, to pave the way for possible future directions.